# An In Vitro Oxidative Stress Model of the Human Inner Ear Using Human-Induced Pluripotent Stem Cell-Derived Otic Progenitor Cells

**DOI:** 10.3390/antiox13111407

**Published:** 2024-11-16

**Authors:** Minjin Jeong, Sho Kurihara, Konstantina M. Stankovic

**Affiliations:** 1Department of Otolaryngology-Head and Neck Surgery, Stanford University School of Medicine, Stanford, CA 94305, USA; jeongm@stanford.edu (M.J.); sho128@stanford.edu (S.K.); 2Department of Otolaryngology-Head and Neck Surgery, The Jikei University School of Medicine, 3-25-8 Nishishimbashi Minato-ku, Tokyo 105-8461, Japan; 3Department of Neurosurgery, Stanford University School of Medicine, Stanford, CA 94305, USA; 4Wu Tsai Neurosciences Institute, Stanford University, Stanford, CA 94305, USA

**Keywords:** cisplatin, gentamicin, human-induced pluripotent stem cells, hydrogen peroxide, otic progenitor cells, oxidative stress

## Abstract

The inner ear organs responsible for hearing (cochlea) and balance (vestibular system) are susceptible to oxidative stress due to the high metabolic demands of their sensorineural cells. Oxidative stress-induced damage to these cells can cause hearing loss or vestibular dysfunction, yet the precise mechanisms remain unclear due to the limitations of animal models and challenges of obtaining living human inner ear tissue. Therefore, we developed an in vitro oxidative stress model of the pre-natal human inner ear using otic progenitor cells (OPCs) derived from human-induced pluripotent stem cells (hiPSCs). OPCs, hiPSCs, and HeLa cells were exposed to hydrogen peroxide or ototoxic drugs (gentamicin and cisplatin) that induce oxidative stress to evaluate subsequent cell viability, cell death, reactive oxygen species (ROS) production, mitochondrial activity, and apoptosis (caspase 3/7 activity). Dose-dependent reductions in OPC cell viability were observed post-exposure, demonstrating their vulnerability to oxidative stress. Notably, gentamicin exposure induced ROS production and cell death in OPCs, but not hiPSCs or HeLa cells. This OPC-based human model effectively simulates oxidative stress conditions in the human inner ear and may be useful for modeling the impact of ototoxicity during early pregnancy or evaluating therapies to prevent cytotoxicity.

## 1. Introduction

Aerobic metabolism by mitochondria is crucial for biological activity, leading to the generation of reactive oxygen species (ROS) as by-products in cells [1]. Under normal physiological conditions, ROS are produced at controlled levels and have essential cellular signaling roles [2]. However, ROS levels rise dramatically during oxidative stress, triggering cellular damage and ultimately leading to cell death, primarily through necrosis or apoptosis [3]. Several cell types in the inner ear, such as hair cells, spiral ganglion neurons, and stria vascularis cells, have high mitochondrial density and are therefore particularly susceptible to oxidative stress [4,5]. As these cells are crucial components for the detection and transmission of sound information by the cochlea, their ROS-mediated damage can result in hearing loss [5,6,7].

Oxidative stress in the auditory system can be triggered by various factors, including noise exposure, aging, ischemia, and ototoxic drugs [5]. Among the latter, aminoglycoside antibiotics and platinum-based anticancer drugs are particularly notorious for their ototoxic effects in clinical practice [8]. For example, the platinum-based chemotherapy agent cis-diamminedichloroplatinum II (cisplatin) is commonly used in the treatment of hematologic malignancies and solid tumors but is highly ototoxic [9]. Approximately 40–60% of patients treated with cisplatin develop some degree of hearing loss [10,11,12], which can occur after just a single cycle [13]. Additionally, cisplatin’s ability to cross the placental barrier raises concerns about its impact on fetal ear development, as highlighted in its prescribing information [14]. Several studies have documented cisplatin’s presence in cord blood and its potential effects on newborns, including hearing loss [15,16,17], although most reviews suggest that platinum-based chemotherapy during the second and third trimesters poses minimal risk to the fetus [18,19]. However, isolated cases, such as severe bilateral perceptive hearing loss in a newborn exposed to cisplatin in utero [20], highlight the need for caution and further investigation.

Although the precise cellular and molecular mechanisms behind cisplatin-induced ototoxicity remain unclear, several potential pathways have been suggested in animal studies, including DNA damage, oxidative stress, and inflammation [21]. Once cisplatin enters cells of the cochlea through passive diffusion or by transporters [22,23], it forms highly reactive complexes that damage DNA, leading to the activation of tumor suppressor p53 [24] and apoptosis through cytochrome c release from mitochondria and caspase 3 activation [25]. Cisplatin generates ROS via NADPH oxidases like NOX3, which are abundant in the cochlea, further promoting cell death [26]. Additionally, cisplatin induces the release of proinflammatory cytokines and activates NF-kB, which exacerbates damage through a cascade of proapoptotic and proinflammatory signals [27,28].

Similarly, gentamicin, an aminoglycoside antibiotic commonly used to treat Gram-negative bacterial infections, also increases ROS production in the inner ear, contributing to ototoxicity with prolonged use or at high dosages [29]. Gentamicin permeates mechanotransducer (MET) channels at the tips of sensory hair cells [30,31] and accumulates in their mitochondria. It then directly inhibits protein synthesis in mitochondrial ribosomes [32,33] and triggers the opening of mitochondria permeability transition pores, leading to mitochondrial dysfunction and the activation of apoptotic pathways [34]. This process, involving cytochrome c release and caspase activation, parallels the mechanism observed with cisplatin.

The incidence of hearing loss following gentamicin treatment varies according to the administration protocol, but up 58% of patients may experience hearing threshold shifts [35]. In vivo animal studies showed that gentamicin causes dose- and time-dependent hair cell damage and significantly elevated auditory brainstem response thresholds [36,37]. Like cisplatin, gentamicin can also cross the placenta [38]. A study on 54 women receiving gentamicin (0.6 to 1.4 mg per kilogram) before cesarean section demonstrated that gentamicin reaches the fetus, with cord blood concentrations being approximately 42% of the simultaneous maternal levels [39]. Although the absolute levels in the cord blood were quite low (<1 µg/mL) one hour after the injection, there are case reports of hearing loss in children due to in utero exposure [40]. Therefore, both gentamicin and cisplatin are generally not recommended during pregnancy unless the benefits clearly outweigh the risks for the mother [14].

Despite the insights gained from animal models or mouse auditory cell lines like HEI-OC1 [41], these systems have limitations when extrapolated to human inner ear biology. Animal models are unable to recapitulate all genes and pathways linked to human hereditary hearing loss and the HEI-OC1 cell line faces limitations such as species differences and artifacts from immortalization [42]. The challenge of obtaining human inner ear tissue, due to its small size and encapsulation within the temporal bone, has driven researchers to seek alternative models. Recently, human pluripotent stem cell-derived otic progenitor cells (OPCs) have emerged as a valuable tool for auditory research [43,44,45]. Although OPCs represent an early stage of inner ear development, they provide a useful platform for studying the human inner ear. Previous studies have demonstrated their utility in modeling congenital hearing loss caused by cytomegalovirus and Zika virus [45], as well as exploring the relationship between SARS-CoV-2 infection and audio-vestibular dysfunction [44].

In this study, we aimed to establish an in vitro oxidative stress model of the human inner ear using OPCs derived from human-induced pluripotent stem cells (hiPSCs) by applying oxidative stressors. Initially, we exposed OPCs to hydrogen peroxide (H_2_O_2_), a ubiquitous type of ROS implicated in oxidative damage in many cell types, including the inner ear where it is linked to hearing disorders [46,47]. This step demonstrated that OPCs undergo oxidative stress-induced damage or cell death in response to H_2_O_2_ exposure. Subsequently, we treated OPCs with gentamicin and cisplatin to evaluate their ototoxic effects by assessing cell viability and cell death. By measuring ROS production, Mito-Tracker staining, cytochrome c location, and caspase 3/7 activity, we investigated whether gentamicin and cisplatin cause oxidative stress and cytotoxicity via mitochondria-dependent apoptosis. Additionally, we compared the oxidative stress responses of OPCs with those of hiPSCs and HeLa cells to better understand the specific vulnerabilities of the human inner ear.

## 2. Materials and Methods

### 2.1. Cell Culture

#### 2.1.1. hiPSCs

The hiPSC line SK8-A, which was generated in our laboratory [44], was used for up to 50 passages. Cells were maintained on Geltrex-coated plates (Gibco, Grand Island, NY, USA, #A1413302) in mTeSR1 medium (StemCell Technologies, Vancouver, BC, Canada, #85850) supplemented with 1× penicillin-streptomycin (Gibco, Grand Island, NY, USA, #15140122). Cells were passaged at approximately 80–90% confluency. ReLeSR (StemCell Technologies, Vancouver, BC, Canada, #100-0483) was applied to detach the hiPSC colonies and the detached cell clumps were transferred to new Geltrex-coated 6-well plates. The medium was replenished daily.

#### 2.1.2. OPC Differentiation from hiPSCs

We followed the published protocol to differentiate hiPSCs into OPCs using a monolayer culture system [44]. Briefly, undifferentiated hiPSCs (SK8-A) were dissociated with ReLeSR and seeded at a density of 30,000 cells/cm^2^ onto laminin-coated plates (R&D Systems, Minneapolis, MN, USA, #3401-010-02).

Days 0 to 13: The differentiation medium was composed of DMEM/F12 (Gibco, Grand Island, NY, USA, #11330032) supplemented with 1× N2 (1% final concentration, Gibco, Grand Island, NY, USA, #17502048), 1× B27 (2% final concentration, Gibco, Grand Island, NY, USA, #17504044), 50 ng/mL fibroblast growth factor (FGF)-3 (R&D Systems, Minneapolis, MN, USA, #1206-F3), and 50 ng/mL FGF-10 (R&D Systems, Minneapolis, MN, USA, #345-FG). The medium was replaced on day 1 and then changed every other day. Additionally, 10 μM Y-27632 (TOCRIS, Bristol, UK, #1254) was included from days 0 to 3 to support cell survival.

Day 14: The cells were detached using TrypLE™ Select Enzyme (Gibco, Grand Island, NY, USA, #12563011) and reseeded onto growth factor-reduced Matrigel (Corning, Corning, New York, USA, #356230)-coated plates at a density of 80,000 cells/cm^2^.

Days 14 to 20: The differentiation medium was DMEM/F12 supplemented with 1× N2, 1× B27, and 5 μM dibenzazepine (DBZ; TOCRIS, Bristol, UK, #4489). The medium also contained 10 μM Y-27632 for days 14. On day 15, the medium was replaced without Y-27632 and the medium was subsequently changed every other day until day 20.

#### 2.1.3. HeLa Cells

HeLa cells were used for up to 20 passages. Cells were maintained in DMEM (Gibco, Grand Island, NY, USA, #10313021) supplemented with 10% fetal bovine serum (FBS; Sigma, St. Louis, MO, USA, #F4135, Lot 23B289), 1× GlutaMAX (Gibco, Grand Island, NY, USA, #35050061), 1× penicillin-streptomycin. Cells were passaged at approximately 80–90% confluency. TrypLE Select (Gibco, Grand Island, NY, USA, #12563029) was applied to detach the cells and pass to new plates. The medium was replenished once every 3 days.

### 2.2. H_2_O_2_, Cisplatin, and Gentamicin Exposure

During the differentiation of hiPSCs into OPCs, cells were transferred on day 14 into 96-well plates for cell viability and propidium iodide (PI) cell death assays, and into 24-well plates for ROS production and caspase 3/7 activity assays. Drug exposure was performed on day 20. For hiPSCs, 10,000 cells were seeded into 96-well plates and cultured for 3 days before cell viability assay. For HeLa cells, 10,000 cells were seeded into 96-well plates and cultured for 1 day before cell viability assay.

H_2_O_2_ (Sigma-Aldrich, St. Louis, MO, USA, #216763) was diluted with distilled water to achieve final concentrations of 1.56, 3.13, 6.25, 12.5, 25, 50, and 100 mM for OPCs and of 15.6, 31.25, 62.5, 125., 250, 500, and 1000 μM for hiPSCs and HeLa cells.

Cisplatin (EMD Millipore, Burlington, MA, USA, #232120) was dissolved in PBS (Gibco, Grand Island, NY, USA, #14080-055) to create a stock solution at 1 mg/mL (approximately 3330 μM) and then further diluted with PBS to obtain final concentrations of 12.5, 25, 50, 100, 200, 400, and 800 μM for OPCs. The final concentrations of cisplatin were 1.56, 3.13, 6.25, 12.5, 25, 50, and 100 μM for hiPSCs and 1.56, 3.13, 6.25, 12.5, 25, 50, 100, and 200 μM for HeLa cells.

Gentamicin (Sigma-Aldrich, St. Louis, MO, USA, #G4918) was dissolved in PBS to prepare a stock solution at 10 mg/mL (approximately 21.6 mM) and subsequently diluted to final concentrations of 75, 150, 300, 600, 1200, 2400, and 4800 μM for all cell lines.

For each experiment, cisplatin and gentamicin were freshly dissolved from powder (H_2_O_2_ is available as a liquid) and all working solutions, including those for H_2_O_2_, were freshly prepared immediately before the experiment. The chosen concentrations were designed to cover a range from 0% to 100% effect, allowing for the determination of half-maximal inhibitory concentration (IC50) values on a logarithmic scale. The vehicle alone served as the negative control.

Cells were exposed to the oxidative agents, Triton-X, or vehicle for 24 h. This timepoint was chosen based on the findings of prior studies documenting the time course of cisplatin exposure in a cell line derived from the inner ear of fetal mice (HEI-OC1) [48,49].

### 2.3. Cell Assays

#### 2.3.1. Cell Viability Assay

At 24 h post-application of H_2_O_2_, cisplatin, or gentamicin, the supernatant was completely removed from the OPCs, hiPSCs, or HeLa cells cultured in 96-well plates. Subsequently, 100 µL of the diluted AquaBluer solution (prepared by mixing 0.1 mL of AquaBluer (Boca Scientific Inc., Dedham, MA, USA, #6015) with 10 mL of culture medium) was added back to each well. AquaBluer solution contains resazurin, a redox indicator that is reduced to resorufin by metabolically active cells. Living cells with active metabolism can reduce resazurin to resorufin via mitochondrial and cytoplasmic enzymes such as NADPH dehydrogenase and flavin reductase.

Wells containing only the medium, without cells, were included to assess background fluorescence. Following 4 h incubation at 37 °C, fluorescent intensity was measured using a SpectraMax iD3 microplate reader (Molecular Devices, San Jose, CA, USA), with an excitation wavelength of 540 nm and emission wavelength of 590 nm. For data analysis, the average background fluorescence (no-cell control) was subtracted from the measured values. Percent viability was then calculated using the following formula: % Viability = (Fluorescence intensity of the test sample/Fluorescence intensity of the vehicle control) × 100. The IC50 calculator provided by AAT Bioquest (https://www.aatbio.com/tools/ic50-calculator (accessed on 12 November 2024)) was used to establish IC50.

#### 2.3.2. Propidium Iodide Cell Death Assay

PI (R&D Systems, Minneapolis, MN, USA, #5135) was diluted to 1:1000 in the culture medium and added to cells in 96-well plates. After applying H_2_O_2_, cisplatin, or gentamicin, real-time imaging was performed every hour for 24 h at 37 °C using an IncuCyte S3 live-cell imaging system (Essen BioScience, Ann Arbor, MI, USA). Specifically, four 20× magnification phase contrast and red fluorescence images were separately captured from each well at 4 different locations. For data presentation in figures, PI-positive cells were quantified by counting red objects per image.

#### 2.3.3. Detection of Caspase 3/7 Enzyme Activity

BioTracker NucView^®^ 530 Red Caspase-3 Dye (Sigma-Aldrich, St. Louis, MO, USA, Cat# SCT105) was utilized to detect caspase 3/7 enzyme activity. After washing the cells cultured in a 24-well plate with PBS, the medium was replaced with fresh medium containing 5 µM NucView^®^ 530 substrate stock solution. Real-time imaging was performed every hour for 24 h at 37 °C using the IncuCyte S3 live-cell imaging system. Specifically, four images at 20× magnification were captured from each well for both phase contrast and red fluorescence images. For fluorescence imaging, the acquisition time was set to 400 ms for the red channel. Image analysis focused on determining the number of red objects per mm^2^, normalized to the 0 h data, using the IncuCyte^®^S3 Software v2018B Basic Analyzer. The threshold for the Red Calibrated Unit (RCU) was set at 0.1.

#### 2.3.4. Detection of ROS Production

CellROX™ Orange Reagent (Thermo Fisher Scientific, Waltham, MA, USA, C10443) was used to detect ROS production. After washing the cells cultured in a 24-well plate with PBS, the medium was replaced with fresh medium containing 5 µM CellROX substrate stock solution, and the cells were incubated for 30 min at 37 °C. After incubation, the medium was removed, and the cells were washed three times with PBS. Real-time imaging was performed every hour for 24 h at 37 °C using the IncuCyte S3 live-cell imaging system. Specifically, four images at 20× magnification were captured from each well for both phase contrast and red fluorescence images. For fluorescence imaging, the acquisition time was set to 400 ms for the red channel. Image analysis focused on determining the average RCU intensity per mm^2^, normalized to the 0 h data, using the IncuCyte^®^S3 Software v2018B Basic Analyzer. The threshold for the RCU was set at 0.1.

### 2.4. Quantitative Reverse Transcription PCR (qRT-PCR)

qRT-PCR was performed on hiPSCs and OPCs (day 20) to detect and quantify transcript levels of the otic lineage-related genes *PAX2, PAX8,* and *GATA3.* Total RNA was isolated from using RNeasy Mini Kits (Qiagen, Hilden, Germany, #74104). For cDNA synthesis, 1 μg of total RNA was used. Reverse transcription was carried out with SuperScript IV (Invitrogen, Thermo Fisher Scientific, Waltham, MA, USA, #18090050) including oligo dT, following the manufacturer’s instructions. qRT-PCR was performed using the QuantStudio 6 Pro Real-Time PCR system (Applied Biosystem, Waltham, MA, USA) with the reaction mixture prepared using GoTaq qPCR Master Mix (Promega, Madison, WI, USA, #A6002).

The primers used were the following: *PAX2* (forward, GACTATGTTCGCCTGGGAGATTC; reverse, AAGGCTGCTGAACTTTGGTCCG; 119 bp), *PAX8* (forward, TCAACCTCCCTATGGACAGCTG; reverse, GAGCCCATTGATGGAGTAGGTG; 137 bp), *GATA3* (forward, ACCACAACCACACTCTGGAGGA; reverse, TCGGTTTCTGGTCTGGATGCCT; 132 bp), *GAPDH* (forward, CTGACTTCAACAGCGACACC; reverse, GTGGTCCAGGGGTCTTACTC).

The PCR cycle parameters were an initial hold at 50 °C for 2 min, followed by 95 °C for 10 min, 40 cycles of denaturation at 95 °C for 15 s, and annealing at 60 °C for 1 min. Expression levels were normalized to that of *GAPDH* and are presented in arbitrary units using the 2^−ΔΔCt^.

### 2.5. Immunocytochemistry

To label mitochondria, 100 nM of Mito-Tracker (Cell Signaling Technology, Danvers, MA, USA, #9082) was added to the cells and incubated at 37 °C for 30 min. After incubation, the following immunostaining procedure was performed.

Cultured cells were washed 3 times with PBS and fixed in 4% paraformaldehyde (Thermo Fisher Scientific, Waltham, MA, USA, # AAJ19943K2) at room temperature for 10 min. For permeabilization, cells were washed 3 times with PBS and incubated in PBST (0.1% Triton X-100 in 1× PBS) at room temperature for 10 min. The cells were blocked with 5% goat serum (Gibco, Grand Island, NY, USA, #PCN5000) or 5% normal horse serum (Abcam, Cambridge, UK, #ab7484) in PBST for 1 h. Subsequently, the cells were incubated overnight at 4 °C with the following primary antibodies diluted in 1% BSA in PBST: goat polyclonal IgG PAX2 (R&D Systems, Minneapolis, MN, USA; #AF3364; 1:50), rabbit polyclonal PAX8 (Abcam, Cambridge, UK; #AB97477; 1:200), rabbit monoclonal GATA3 (Cell Signaling Technology, Danvers, MA, USA; #5852; 1:800), and mouse monoclonal cytochrome c (Cell Signaling Technology, Danvers, MA, USA; #12963; 1:300).

The next day, the cells were washed three times with PBS and incubated with the following secondary antibodies in 1% BSA in PBST: chicken anti-goat IgG 488 (Invitrogen, Thermo Fisher Scientific, Waltham, MA, USA; #A21467; 1:500) or goat anti-rabbit IgG 488 (Invitrogen, Thermo Fisher Scientific, Waltham, MA, USA; #A11008; 1:1000). The nuclei were stained with 0.1 μg/mL DAPI (Cell Signaling Technology, Danvers, MA, USA, #4083) and Vectashield (Vector Laboratories, Burlingame, CA, USA, # H1000) was used to mount the samples. The samples were analyzed using ZEISS LSM 880 confocal microscope (ZEISS, Oberkochen, Germany).

### 2.6. Statistical Analysis and Reproducibility

Statistical analyses were performed using Prism 10.3. A *p*-value of ≤0.05 was considered statistically significant for all analyses. For qPCR and cell viability assays, unpaired t-tests were used to compare gene expression between hiPSCs and OPCs, as well as to evaluate differences between control samples and drug-exposed samples, respectively. For cell death assays, multiple unpaired t-test was used for comparison between control and drug exposed samples.

A repeated measures two-way ANOVA with Geisser–Greenhouse correction was performed to assess the effects of time and treatment on detection of caspase 3/7 enzyme activity and ROS production. The assumption of sphericity was violated, as assessed by Mauchly’s test of sphericity, and therefore the Geisser–Greenhouse correction was applied to adjust the degrees of freedom. Post-hoc analysis was performed using Tukey’s multiple comparisons test at each time point to determine specific group differences. Two-way ANOVA with Geisser–Greenhouse correction was used for quantification of active mitochondria.

Representative bright-field images in Figure 1b show the results of at least 10 different passages of hiPSCs (SK8-A line) and their subsequent differentiation into OPCs. Bar graphs in Figure 1c display results from 3 biological replicates and 2–3 technical replicates for each time point. Immunostaining images shown in Figure 1d are representative of a minimum of three independent experiments. The cell viability, cell death, caspase 3/7, and ROS production assay in Figure 2a, Figure 2b, Figure 3a, Figure 3b, Figure 4a, Figure 4b, Figure 5b and Figure 5d, are based on the combined results of at least 3 independent experiments, each with 3–4 technical replicates. The representative bright-field together with red fluorescent PI images in Figure 2c, Figure 3c and Figure 4c represent results from at least 3 independent experiments, each with 3 technical replicates. Immunostaining images shown in Figure 6a,b are representative of a minimum of three independent experiments. For quantification of active mitochondria presented in Figure 6c,d, at least 3 images from each of 3 independent experiments were analyzed. The cell viability, cell death, caspase 3/7, and ROS production assay in Figure 7b,d are based on the combined results of at least 3 independent experiments, each with 3–4 technical replicates. Additionally, the representative bright-field combined with red fluorescent caspase 3/7 images in Figure 5a,c and ROS images in Figure 7a,c represent 3 independent experiments, each with 4 technical replicates.

## 3. Results

### 3.1. Generation of OPCs from hiPSCs to Model Embryonic Human Inner Ear Cells

FGF3, FGF10, and the Notch inhibitor DBZ were applied to SK8-A hiPSCs to generate OPCs as previously described [43,44,45] (Figure 1a). After 20 days of differentiation, hiPSC colonies developed into individual OPCs progressing through the placodal stage (Figure 1b).

qRT-PCR analysis of OPCs at day 20 revealed significant upregulation of otic lineage-related genes *PAX2* (280.05 ± 139.12-fold increase), *PAX8* (218.88 ± 107.46), and *GATA3* (816.69 ± 138.89) compared to undifferentiated hiPSCs (all *p* < 0.05) (Figure 1c). Further identification of OPCs was confirmed through immunostaining, with 94.39 ± 1.45% of cells expressing PAX2, 98.14 ± 0.45% PAX8, and 97.27 ± 3.03% GATA3 (Figure 1d). Together, these findings indicate that the hiPSCs successfully differentiated to OPCs resembling the otocyst stage in vivo, which typically forms at approximately days 27–28 of human embryonic development [50].

### 3.2. Effects of H_2_O_2_ Exposure

To confirm the effects associated with an established oxidative agent, OPCs (day 20), hiPSCs, HeLa cells were exposed to varying concentrations of H_2_O_2_ for 24 h, then cell viability was assessed with AquaBluer. A significant, dose-dependent reduction in OPC viability was observed starting at 25.0 mM H_2_O_2_ (IC50 of 24.1 ± 2.89 mM) in comparison with vehicle-exposed OPCs (*p* = 0.0443) (Figure 2a). hiPSCs and HeLa cells demonstrated greater vulnerability to H_2_O_2_ than OPCs, and cell viability was significantly reduced from 62.5 μM (*p* = 0.020) and 500 μM (*p* = 0.029), with IC50 values of 129.20 ± 10.59 μM and 373.66 ± 164.42 μM, respectively.

The rate and extent of OPC death over 24 h of H_2_O_2_ exposure were assessed using PI in a real-time live-cell imaging system. As membrane integrity is compromised in dying cells, PI enters and fluorescently labels the nuclei [51]. Concentrations of approximate IC50 (25 mM) and half of IC50 (12.5 mM) H_2_O_2_ in PI-containing media were selected for further testing with OPCs; the number of PI-positive cells was quantified hourly. The results demonstrated a gradual increase in the number of red PI-positive (dead) cells in both the 12.5 mM and 25 mM H_2_O_2_ groups, while the control group remained relatively stable (Figure 2b). Both 12.5 mM and 25 mM H_2_O_2_-treated groups had significantly more dead cells compared to the control group between 8 to 18 h and 2 to 24 h of H_2_O_2_ exposure, respectively. Additionally, widespread clumping of red-fluorescing cells, indicating DNA bound by PI, was observed 24 h post-exposure to 12.5 mM and 25 mM H_2_O_2_, with notably fewer occurrences in control cells (Figure 2c). These results demonstrate that the oxidative stressor H_2_O_2_ prompts damage and cell death in OPCs.

### 3.3. Effects of Gentamicin Exposure

To evaluate the ototoxic effects of gentamicin on OPCs, day 20 cells were exposed from concentrations ranging from 0 to 4800 μM for 24 h. OPC viability was significantly reduced starting at 150 μM gentamycin compared to control (*p* = 0.0032) (Figure 3a). The reduction in OPC viability was dose-dependent and the IC50 of 472.17 ± 75.21 μM is consistent with the reported range affecting outer hair cell survival in rat cochlea explants (563 ± 28 μM) [52].

Interestingly, gentamicin exposure did not affect the viability of either hiPSCs or HeLa cells. Even at the highest gentamicin concentration (4800 μM), the viability of hiPSCs and HeLa cells were 98.48 ± 8.41% and 99.55 ± 5.83%, respectively, a concentration at which only 26.04 ± 4.22% of OPCs survived. hiPSCs had numerically, but not significantly, higher average cell viability across the 75 to 2400 μM range compared to the vehicle-treated group. The viability of HeLa cells also increased within this concentration range and were significantly higher compared to the vehicle-treated group at 150 μM, 300 μM, and 600 μM, albeit with minor increases of only 2.38% to 3.92%.

OPC death quantified by number of PI-positive cells revealed an overall increasing trend following gentamicin exposure. The number of PI-positive cells was significantly higher than controls during 13 to 24 h for the 150 μM group (*p* < 0.05) and 14 to 24 h for the 600 μM group (both *p* < 0.05) (Figure 3b). By the end of the 24 h exposure, both the 150 μM (*p* = 0.002) and 600 μM (*p* < 0.001) gentamicin-treated groups had significantly more red PI-positive cells than the vehicle-treated group (Figure 3c), although no significant difference was observed between the two gentamicin concentrations. In addition, real-time phase contrast measurements of cell confluency indicated a statistically significant difference between the exposure groups only at 23 to 24 h (*p* < 0.05, Appendix A). However, the overall trend of dose-dependent effects was more pronounced in the confluency measurements compared to the PI assays. These findings revealed that OPCs have heightened vulnerability to gentamicin-induced cytotoxicity compared with hiPSCs and HeLa cells.

### 3.4. Effects of Cisplatin Exposure

To assess the cytotoxic effects of cisplatin, day 20 OPCs were exposed to concentrations ranging from 0 to 800 μM for 24 h. OPC viability was significantly reduced starting at 12.5 μM cisplatin compared to vehicle-exposed OPCs (*p* = 0.018) (Figure 4a). This reduction in OPC viability was dose-dependent, with an IC50 of 57.10 ± 16.46 μM. Similar to the findings of the H_2_O_2_ exposure experiments, hiPSCs and HeLa cells were more sensitive to cisplatin than OPCs. Both cell types showed significantly decreased viability with 3.13 μM (*p* = 0.0434) and 12.5 μM (*p* = 0.0013) cisplatin, with IC50 values of 2.40 ± 1.57 μM and 25.50 ± 3.6 μM, respectively. The IC50 for OPCs was higher than that for HeLa cells; however, their effective ranges overlapped.

While gentamicin induced rapid cell death as early as one hour after exposure, cisplatin-induced cell death, as measured by PI staining, only appeared after 11 h for 25 μM cisplatin and 10 h for 50 μM cisplatin (both *p* < 0.05) (Figure 4b). By the end of the 24 h exposure, both the 25 μM (*p* = 0.003) and 50 μM (*p* < 0.001) cisplatin-treated groups had significantly more PI-positive cells compared to the vehicle-treated group (Figure 4c). The delayed yet significant cytotoxic effect of cisplatin on OPCs suggests that the accumulation of cisplatin in cells over time is necessary for the ototoxic effects.

### 3.5. Apoptosis in OPCs

Among the various pathways triggered by oxidative stress (i.e., apoptosis, necrosis, necroptosis, and ferroptosis [53,54]), apoptosis is the predominant form of cell death in the inner ear when exposed to gentamicin [34,55] and cisplatin [56,57]. Therefore, we evaluated caspase 3/7 activity, a key indicator of apoptotic cell death, to quantify apoptosis in OPCs treated with gentamicin and cisplatin.

First, caspase 3/7 activity was measured in OPCs following treatment with 300 μM and 600 μM gentamicin compared to vehicle control (Figure 5a,b). A significant interaction between time (hours) and gentamicin treatment was observed (F(48, 792) = 5.594, *p* < 0.0001), indicating that the changes in enzyme activity over time differed between the treatment groups. Significant main effects of time (F(1.221, 40.28) = 235.5, *p* < 0.0001) and treatment (F(2, 33) = 5.975, *p* = 0.0061) were observed, demonstrating that gentamicin impacted caspase 3/7 activity. Post-hoc analysis revealed that 600 μM gentamicin exposure induced a significantly stronger apoptotic response compared to control from 5 to 24 h. No significant differences were observed in caspase 3/7 activity between the 300 μM gentamicin and control group at any time point, although it was numerically higher in the 300 μM group between 5 and 24 h of exposure.

Caspase 3/7 activity was also assessed in OPCs treated with vehicle, 25 μM, and 50 μM cisplatin (Figure 5c,d). There was a significant interaction between time and cisplatin treatment (F(48, 792) = 5.329, *p* < 0.0001), as well as significant main effects of time (F(1.150, 37.94) = 291.1, *p* < 0.0001) and treatment (F(2, 33) = 3.459, *p* = 0.0433). Post-hoc analysis showed that 50 μM cisplatin induced significantly higher caspase 3/7 activity compared to control during 11 to 24 h of exposure (*p* < 0.05). There were no significant differences in caspase 3/7 activity between the 25 μM cisplatin and control groups except at 24 h (*p* = 0.04), although the average caspase 3/7 activity was numerically higher than the control throughout the experiment.

When investigating the mechanisms of cell death induced by cisplatin and gentamicin, we observed a significant increase in caspase 3/7 activity at concentrations near the IC50. However, at lower concentrations (150 μM gentamycin and 12.5 μM cisplatin), although cell viability was significantly reduced (Figure 3a and Figure 4a), we did not detect a significant increase in caspase 3/7 activity. This suggests the involvement of non-apoptotic mechanisms, such as ferroptosis, in cell death triggered by these drugs. Taken together, these findings suggest that both gentamicin and cisplatin induce apoptosis in OPCs, with higher concentrations resulting in a more robust apoptotic response.

### 3.6. Effects on Mitochondria in OPCs

To assess whether the elevation of caspase 3/7 activity caused by gentamicin and cisplatin in OPCs is linked to mitochondrial-dependent apoptosis, we used the Mito-Tracker fluorescent probe to evaluate changes in mitochondrial transmembrane potential following drug exposure. Mito-Tracker staining showed that, prior to exposure (0 h), mitochondria were distributed relatively homogeneously around the cell nucleus (Figure 6a,b). However, after 6 and 24 h of treatment with 600 μM gentamicin or 50 μM cisplatin, this homogenous mitochondrial distribution was disrupted.

To quantify the ratio of active mitochondria, we measured the percentage of Mito-Tracker positive area and normalized it by the number of DAPI-stained nuclei at each time point with different drug concentrations. Following 6 and 24 h of exposure with 300 μM gentamicin, mitochondrial signal levels were reduced to 87.54 ± 5.21% (non-significant, *p* = 0.096) and 37.04 ± 14.88% (*p* = 0.033), respectively, compared to control (0 h) (Figure 6c). At 600 μM gentamicin, the reduction was more pronounced, with mitochondrial signal at 52.58 ± 3.18% (*p* = 0.002) and 13.51 ± 3.43% (*p* < 0.0001) at 6 and 24 h, respectively. Cisplatin produced similar effects: following 25 μM exposure for 6 and 24 h, mitochondrial signal dropped to 69.38 ± 6.59% (*p* = 0.027) and 22.12 ± 3.09% (*p* < 0.0001), respectively (Figure 6d). At 50 μM cisplatin, mitochondrial expression decreased further to 42.29 ± 8.18% (*p* = 0.012) and 9.63 ± 1.56% (*p* < 0.0001) at the 6 and 24 h time points, respectively.

Cytochrome c is a protein located in the mitochondria, crucial for the electron transport chain, which is involved in cellular respiration and energy production [58]. During apoptosis, cytochrome c is released from mitochondria into the cytosol where it helps activate caspases, thereby triggering the apoptotic pathway [58]. At both the 6 and 24 h time points post-treatment, we observed cytochrome c expression throughout the cytoplasm, indicated by its location outside the regions marked by Mito-Tracker staining (Figure 6a,b). Together, the reduction in Mito-Tracker staining, which reflects a loss of mitochondrial membrane potential, along with the release of cytochrome c into the cytosol, suggests that gentamicin and cisplatin induce mitochondrial dysfunction leading to OPC apoptosis.

### 3.7. Measurement of ROS Production in OPCs

To further explore the mechanisms behind this mitochondrial damage, we investigated whether oxidative stress via ROS production is involved in gentamycin and cisplatin-mediated cytotoxicity.

We examined whether ROS production was elevated following gentamicin administration using the CellROX reagent through live-cell imaging (Figure 7a). Because low levels of ROS are present in healthy cells as by-products of normal metabolism [59], we measured ROS intensity instead of counting number cells that express ROS. The mean intensity was plotted at 1 h intervals following the addition of vehicle, 300 μM gentamicin, or 600 μM gentamicin. There was a significant interaction between time and treatment (F(48, 792) = 12.45, *p* < 0.0001), accounting for 11.72% of the total variation. This indicates that the changes in ROS production over time differed significantly between time and treatment (F(48, 792) = 3.215, *p* < 0.0001), accounting for 3.39% of the total variation. A significant main effect of time was also observed (F(1.529, 50.46) = 34.16, *p* < 0.0001), with Geisser–Greenhouse’s epsilon correction applied (ε = 0.06371), contributing 18.02% of the total variation. Additionally, there was a significant main effect of treatment (F(2, 33) = 6.635, *p* = 0.0038), accounting for 17.55% of the total variation. Subject variability between the 300 and 600 μM gentamicin groups accounted for 43.64% of the total variation and was also significant (*p* < 0.0001). These findings demonstrate that both time and treatment had significant effects on ROS production, with significant differences observed between treatment groups over time. Further, gentamicin exposure induced significantly greater mean ROS intensity compared to control in OPCs, beginning at 6 h for 600 μM and at 8 h for 300 μM (both *p* < 0.05) (Figure 7b).

ROS are similarly implicated as a cause of cisplatin-induced cytotoxicity in the inner ear. Therefore, we measured mean ROS intensity in OPCs’ cytoplasm at 1 h intervals following exposure to vehicle, 25 μM cisplatin, or 50 μM cisplatin. (Figure 7c). There was a significant interaction between time and treatment (F(48, 792) = 16.18, *p* < 0.0001), accounting for 13.58% of the total variation among the control, 25 μM cisplatin, and 50 μM cisplatin groups. Furthermore, a significant main effect of time was observed (F(1.824, 60.21) = 72.26, *p* < 0.0001), with Geisser–Greenhouse’s epsilon correction applied (ε = 0.076), contributing 30.32% of the total variation. There was also a significant main effect of treatment (F(2, 33) = 27.14, *p* < 0.0001), accounting for 26.28% of the total variation. These findings demonstrate that both time and treatment had significant effects on ROS production, with notable differences observed between treatment groups over time. The results also suggest that the cytotoxic effects of cisplatin on OPCs are associated with ROS production, with higher concentrations of cisplatin leading to increased ROS levels over the time course studied.

Cisplatin exposure induced significantly greater mean ROS intensity compared to control in OPCs, starting at 8 h with 50 μM cisplatin (*p* = 0.014) and 17 h with 25 μM (*p* < 0.05), and the difference increased over time (Figure 7d). Additionally, mean ROS intensity was higher in 50 μM versus 20 μM cisplatin from 8 to 24 h (*p* = 0.007). Between the 15 h and 24 h time points, the difference in ROS production between the vehicle and cisplatin 50 μM groups was highly significant (*p* < 0.0001). These results indicate that cisplatin, particularly at 50 μM, significantly increased ROS production over time and in a dose-dependent manner compared to control.

## 4. Discussion

To the best of our knowledge, this study provides the first evidence establishing an in vitro oxidative stress model for the human inner ear using OPCs treated with H_2_O_2_, gentamicin, and cisplatin. Our findings highlight the physiological relevance of OPCs as a model for studying oxidative stress in the inner ear. Notably, while the IC50 for gentamicin-treated HEI-OC1 cells is reported at 2 mM [60], our OPCs exhibited an IC50 of 472.17 ± 75.21 μM. This value aligns with the reported IC50 of gentamicin affecting hair cells in rat cochlea explants, which is approximately 563 ± 28 μM [52]. The vulnerability of OPCs to gentamicin exposure suggests their suitability as an in vitro model for the human inner ear, especially when compared to HeLa cells and hiPSCs, which remained unaffected by gentamicin concentrations up to 4800 μM.

Although further studies are required to fully elucidate the mechanisms responsible for gentamicin entry in OPCs and to rule out alternative pathways such as endocytosis, the specific response observed in OPCs suggests the potential presence of MET, transient receptor potential, or adenosine triphosphate receptor channels—channels through which aminoglycosides typically enter hair cells [61]. In contrast, hiPSCs and HeLa cells may lack these channels, or their insensitivity to gentamicin could be due to factors like less permeable membranes, active efflux pumps, or robust protective mechanisms.

Interestingly, although we noted that OPC death began at the lowest tested gentamicin concentration of 75 μM, approximately 26% of OPCs remained viable even after 24 h of exposure to 4800 μM gentamicin. Given the cytotoxicity of gentamicin observed between 50 μM and 1 mM in mouse inner ear studies [62,63,64], a concentration of 4800 μM would typically lead to near-complete cell death. This observation suggests that OPCs may not be a homogeneous cell type; potentially due to incomplete differentiation among the cells or the presence of distinct subpopulations of OPCs with differing susceptibilities. For example, the survival of a subset of OPCs could be due to the absence of MET channels, similar to what has been observed in developing mice [65,66]. In these models, the fluorescent intensity of gentamicin-conjugated Texas Red (GTTR) gradually increases in hair cells from postnatal day 0 (P0) to P4 [65] as MET channels are first detected and become functional [66].

While this model provides valuable insights, it does not fully replicate clinical scenarios where gentamicin-induced ototoxicity typically manifests after days or even months [67]. This delayed onset is likely due to complex physiological processes, such as autophagic flux [67] or the intricate pathways through which gentamicin enters the cochlea and its cells [61]. Systemically administered aminoglycosides first reach the capillaries in the stria vascularis [68], where they interact with various cell subtypes [69], cross the blood–labyrinth barrier (the initial site of entry from the bloodstream into the cochlea) [68], and eventually reach the endolymph [70]. Additionally, there is evidence suggesting alternative routes from the perilymph to the endolymph via Reissner’s or basilar membranes [71,72,73]. These in vivo environments involve dynamic interactions between multiple cell types, and systemic factors like metabolism and immune response are difficult to mimic in vitro. In isolated in vitro cell culture systems, including those using highly proliferative cell lines like HeLa and hiPSCs, the challenges of replicating prolonged exposure and slower progression are even greater. Rapid cell proliferation leads to early confluence, limiting the duration of culture and preventing the observation of delayed effects.

Our findings also demonstrate that hiPSCs are more sensitive to both H_2_O_2_ and cisplatin compared to HeLa cells. Previous research indicates that human embryonic stem cells maintain stringent genomic integrity by enhancing ROS removal capacity and limiting ROS production due to their small mitochondrial load [74]. Moreover, to preserve their pluripotent state, hiPSCs and embryonic stem cells have evolved robust defense mechanisms to tightly regulate ROS levels compared to differentiated cells (in our case OPCs) [75]. In contrast, HeLa cells, as cancer cells, have adapted to tolerate higher ROS levels [76], prioritizing continuous proliferation over genomic integrity. As a result, HeLa cells exhibit greater resilience to oxidative stress compared to hiPSCs, reflecting the fundamental biological differences between stem cells and cancer cells.

Utilizing OPCs for otologic studies offers several advantages. First, OPCs are derived from hiPSCs, which can propagate indefinitely, providing an unlimited supply of cells for experiments. This characteristic is crucial for high-throughput screening and reproducibility, making OPCs a scalable and reliable resource for auditory research. Importantly, while OPCs represent an early and relatively immature stage of inner ear development, they provide practical benefits over other inner ear cell types derived from hiPSCs, such as inner ear organoids [77]. OPCs can be produced relatively quickly and cost-effectively with minimal growth factor requirements, making them accessible for a wide range of studies. Their efficiency does not compromise relevance; OPCs have been successfully employed to model congenital hearing loss [45] and SARS-CoV-2 infections in the human inner ear, demonstrating their utility in studying complex auditory disorders [44].

In conclusion, our findings highlight the potential of using hiPSC-derived OPCs to model the effects of oxidative stress caused by ototoxic drugs in the human inner ear. This model could be a valuable resource for auditory and pharmacology researchers given the substantial challenges of obtaining living human cochlear tissue via biopsy or post-mortem. Potential applications could include detailed interrogations of cell-, gene-, and drug-based auditory disorders and development of therapeutic strategies aimed at mitigating cell death and preserving auditory function. Future studies should focus on elucidating the specific molecular pathways involved in OPC susceptibility to oxidative stress and ototoxicity, which could pave the way for developing targeted interventions to protect hearing.

## Figures and Tables

**Figure 1 antioxidants-13-01407-f001:**
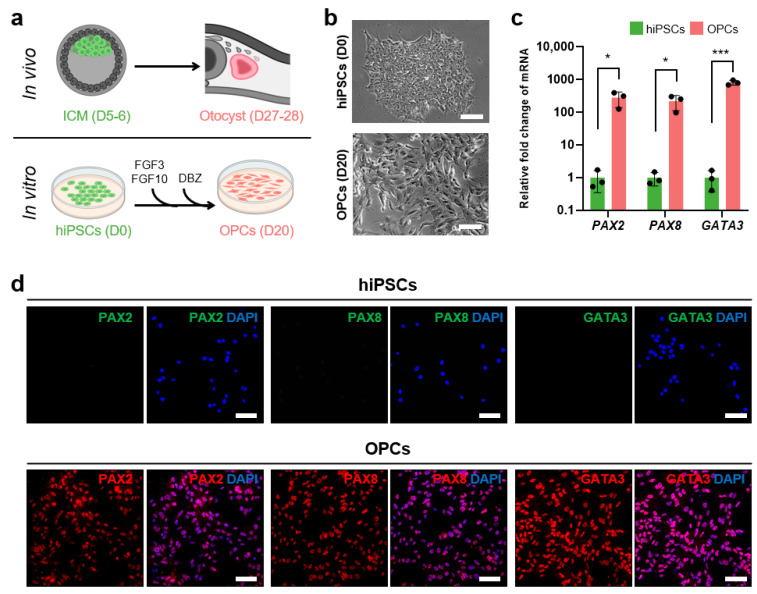
Derivation and characterization of OPCs from hiPSCs. (**a**) Scheme illustrating the generation of OPCs from hiPSCs and corresponding stages of human development. hiPSCs resemble the inner cell mass (ICM), while OPCs correspond to the otocyst stage. (**b**) Representative bright-field images of SK8-A hiPSCs on day 0 and derived OPCs on day 20. Scale bars, 100 µm. (**c**) qRT-PCR data showing fold changes in mRNA expression of otic-related markers (*PAX2*, *PAX8*, and *GATA3*) in OPCs compared to hiPSCs. The expression levels in hiPSCs were set to 1. Mean ± SD; * *p* < 0.05; *** *p* < 0.001. (**d**) Representative immunocytochemistry images showing protein expression of otic lineage markers in OPCs. hiPSCs were used as negative controls to verify the absence of false-positive signals. Scale bars, 100 µm. Abbreviations: *PAX2*, paired box 2; *PAX8*, paired box 8; *GATA3*, GATA binding protein 3. See “Statistical analysis and reproducibility” in Section 2.6 for statistics and experimental information.

**Figure 2 antioxidants-13-01407-f002:**
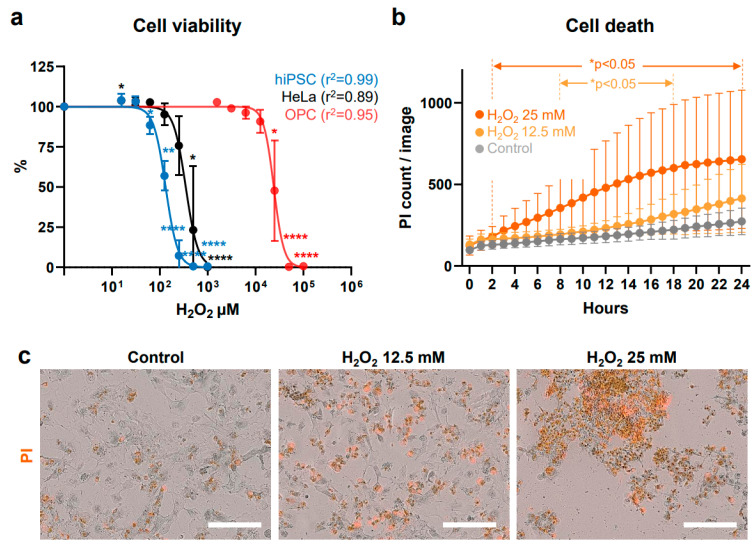
Effects of H_2_O_2_ on cell viability and death. (**a**) Dose-response curves of H_2_O_2_ in hiPSCs, HeLa cells, and OPCs. The percentages of viable hiPSCs and HeLa cells were measured after treatment with 15.625, 31.25, 62.5, 125, 250, 500, and 1000 μM H_2_O_2_ for 24 h. OPCs were treated with 1.56, 3.13, 6.25, 12.5, 25, 50, and 100 mM H_2_O_2_ for 24 h. * *p* < 0.05, ** *p* < 0.01, **** *p* < 0.0001. (**b**) Real-time cell death measurements under control (vehicle), 12.5 mM, and 25 mM H_2_O_2_ conditions are displayed as gray, light orange, and orange circles, respectively. Cell viability is expressed as the mean percentage ± SD. (**c**) Phase contrast and merged red fluorescent propidium iodide (PI)-positive images of OPCs, taken 24 h post-exposure to H_2_O_2_. Images were captured at 20× magnification using the IncuCyte^®^ imager. Scale bars, 200 μm. See “Statistical analysis and reproducibility” in Section 2.6 for statistics and experimental information.

**Figure 3 antioxidants-13-01407-f003:**
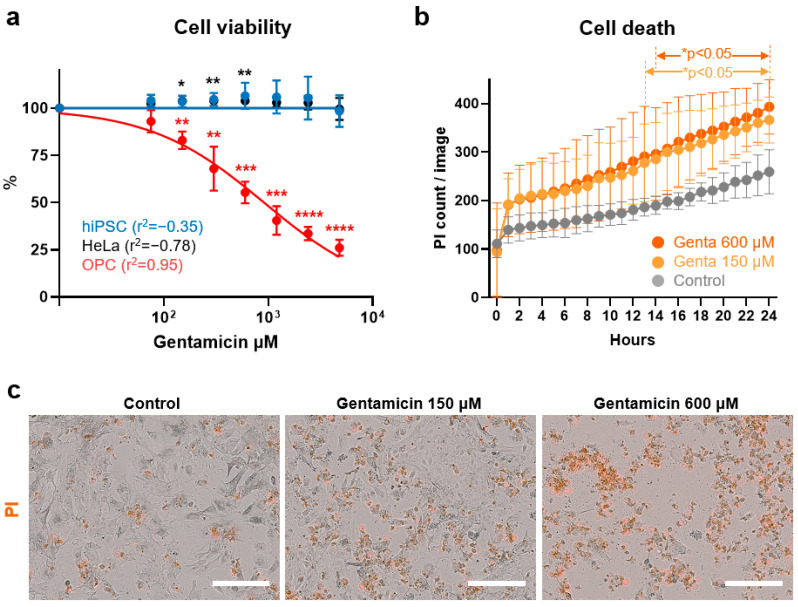
Effects of gentamicin on cell viability and death. (**a**) Dose-response curves of the gentamicin in hiPSCs, HeLa cells, and OPCs. The percentage of viable hiPSCs, HeLa cells, and OPCs were measured after treatment with 75, 150, 300, 600, 1200, 2400, and 4800 μM gentamicin for 24 h. Cell viability is expressed as the mean percentage ± SD. * *p* < 0.05, ** *p* < 0.01, *** *p* < 0.005, **** *p* < 0.0001. (**b**) Real-time cell death measurements under 0 (vehicle), 150, and 600 μM gentamicin (genta) conditions are displayed as gray, light orange, and orange circles, respectively. (**c**) Phase contrast and merged red fluorescent propidium iodide (PI)-positive images of OPCs, taken 24 h post-treatment with gentamicin. Images were captured at 20× magnification using the IncuCyte^®^ imager. Scale bars, 200 μm. See “Statistical analysis and reproducibility” in Section 2.6 for statistics and experimental information.

**Figure 4 antioxidants-13-01407-f004:**
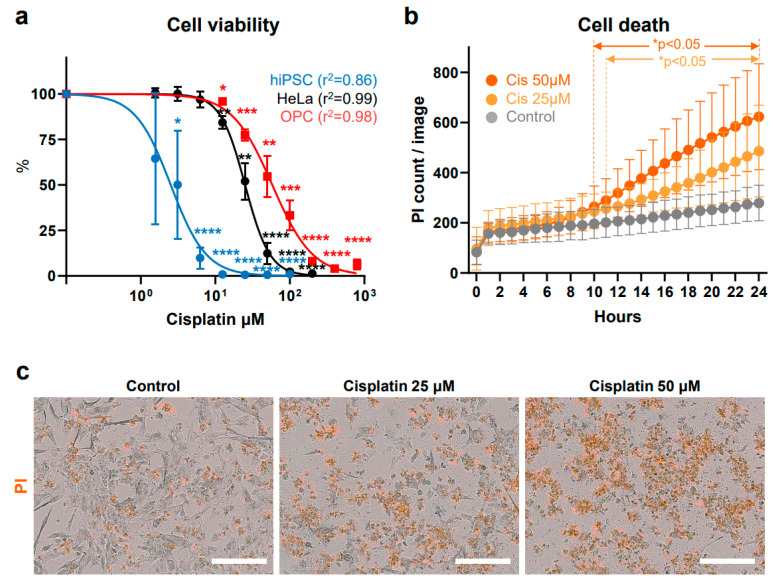
Effects of cisplatin on cell viability and death. (**a**) Dose-response curves of the cisplatin in hiPSCs, HeLa cells, and OPCs. Cell viability of hiPSCs was measured after treatment with 1.56, 3.12, 6.25, 12.5, 25, 50, and 100 μM cisplatin for 24 h. Cell viability of HeLa cells was measured after treatment with 1.56, 3.13, 6.25, 12.5, 25, 50, 100, and 200 μM cisplatin for 24 h. OPCs were treated with 12.5, 25, 50, 100, 200, 400, and 800 μM cisplatin for 24 h. Cell viability is expressed as the mean percentage ± SD. * *p* < 0.05, ** *p* < 0.01, *** *p* < 0.005, **** *p* < 0.0001. (**b**) Real-time cell death measurements under 0 (vehicle), 25 and 50 μM cisplatin (cis) conditions are displayed as gray, light orange, and orange circles, respectively. (**c**) Phase contrast and merged red fluorescent propidium iodide (PI)-positive images of OPCs, taken 24 h post-treatment with cisplatin. Images were captured at 20× magnification using the IncuCyte^®^ imager. Scale bars, 200 μm. See “Statistical analysis and reproducibility” in Section 2.6 for statistics and experimental information.

**Figure 5 antioxidants-13-01407-f005:**
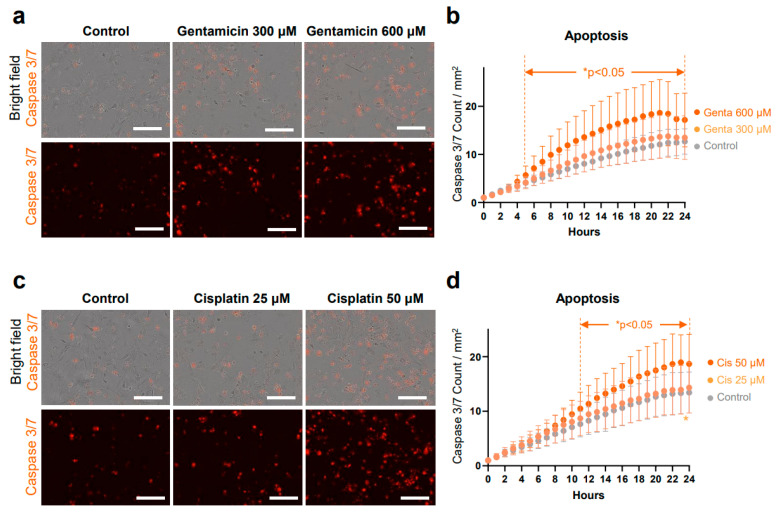
Real-time detection of apoptosis in live OPCs. Apoptosis was visualized using BioTracker NucView caspase 3/7 enzyme activity as a red fluorescent signal. (**a**) OPCs were treated with vehicle, 300, and 600 μM gentamicin and observed over 24 h. Representative images of OPCs 20 h post-treatment. Top row shows merged phase contrast and red fluorescent images, bottom row shows red fluorescent images. (**b**) Red object count plotted over time, normalized to 0 h. A significant increase in caspase 3/7 enzyme activity was observed in the 600 μM gentamicin (genta) group compared to the vehicle group from 5 h. (**c**) Representative images of OPCs 20 h post-treatment with vehicle, 25, and 50 μM cisplatin. (**d**) Red object count plotted over time, normalized to 0 h. From 11 h, a significant increase in caspase 3/7 enzyme activity was observed in the 50 μM cisplatin (cis) group compared to the vehicle group. Data represent the mean of 3 independent experiments, each with 4 technical replicates per time point ± SD. Scale bars, 200 μm.

**Figure 6 antioxidants-13-01407-f006:**
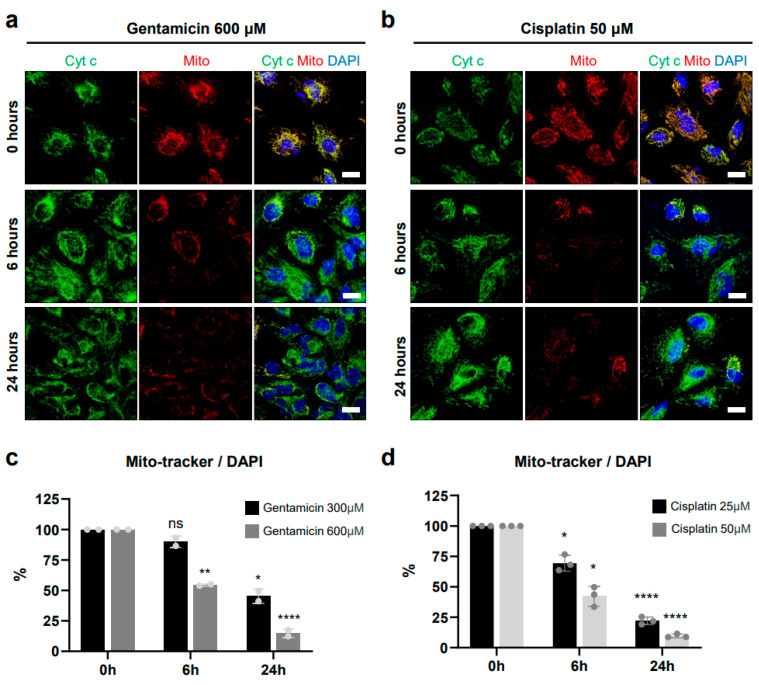
Effects of gentamicin and cisplatin on OPC mitochondria. (**a**,**b**) Representative fluorescence images of OPCs stained with DAPI (nuclei in blue), Mito-Tracker (mitochondria in red), and cytochrome c (cyt c in red) at 0, 6, and 24 h after treatment with 600 μM gentamicin (**a**) and 50 μM cisplatin (**b**). Scale bars: 20 μm. (**c**,**d**) Quantification of active mitochondria by measuring the percentage of Mito-Tracker positive area, normalized to the number of DAPI-stained nuclei at 0, 6, and 24 h after treatment with 300 and 600 μM gentamicin (**c**), and 25 and 50 μM cisplatin (**d**). Mito-Tracker positivity at 0 h was set to 100%. ns, not significant, * *p* < 0.05, ** *p* < 0.01, **** *p* < 0.0001. See “Statistical analysis and reproducibility” in Section 2.6 for statistics and experimental information.

**Figure 7 antioxidants-13-01407-f007:**
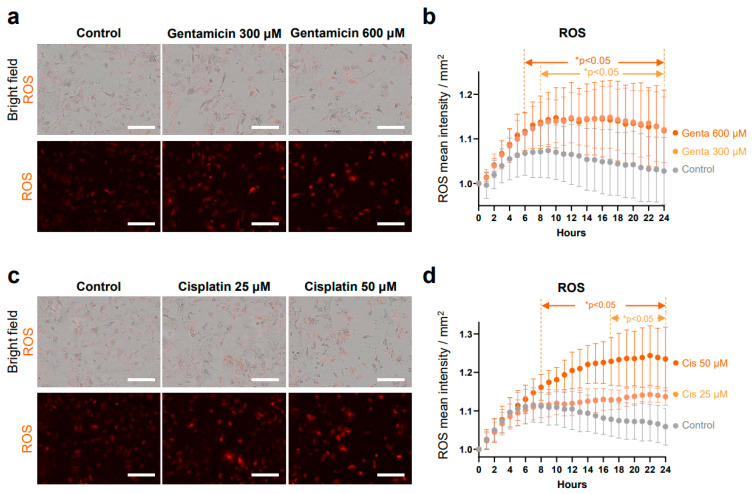
Real-time detection of reactive oxygen species (ROS) in live OPCs. (**a**) Intracellular ROS production was visualized using CellROX reagent as a red fluorescent signal. OPCs were treated with vehicle, 300, and 600 μM gentamicin and observed over 24 h. Representative images of OPCs 12 h post-treatment are displayed. The top row shows merged phase contrast and red fluorescent images, while the bottom row shows red fluorescent images alone. (**b**) Red mean intensity was plotted over time, normalized to 0 h. A significant increase in ROS production was observed in the 600 μM gentamicin (genta) group compared to the vehicle group from 6 h, and in the 300 μM gentamicin group from 8 h. (**c**) Representative images of OPCs 12 h post-treatment with vehicle, 25, and 50 μM cisplatin. (**d**) Red mean intensity plotted over time, normalized to 0 h. From 8 h, a significant increase in ROS production was observed in the 50 μM cisplatin (cis) group compared to the vehicle group. From 16 h, a significant increase was also observed in the 25 μM cisplatin group compared to the vehicle group. Data represent the mean of 3 independent experiments, each with 4 technical replicates per time point ± SD. * *p* < 0.05. Scale bars, 200 μm.

## Data Availability

All source data supporting this study are available within the paper. Additional data are available from the corresponding author upon reasonable request.

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
