# Peer review of "An In Vitro Oxidative Stress Model of the Human Inner Ear Using Human-Induced Pluripotent Stem Cell-Derived Otic Progenitor Cells"

_antioxidants, 2024, doi:10.3390/antiox13111407_

Round 1

Reviewer 1 Report

Jeong et al. showed that OPC is the potential model mimicking human inner ear and useful for oxidative stress evaluation in challenging cisplatin and gentamicin. Overall, the paper is well written and scientifically succinct. Several my major concerns are as follows.

1.  OPC is characterized by the expression of Pax2, Pax8, and Gata3.  In figure 1, the immunostaining requires negative control to check false-positive signals. hiPSC or Hela cell line can be used for negative control.

2. In H2O2 tox experiments, it's a pity that the authors did not use expoential scale doses, which is typically used for PK profiling. It would be better if 12.5 mM and 37.5mM or 125mM had been used instead of 12.5 and 25 mM doses.

3. I wonder if OPC expresses MET channel, which is the channel for aminoglycoside transport. As far as I know, it does not feel that it expresses TMC channel. If not, how could the authors explain the influx and accumulation of gentamicin inside the OPC cells? More detailed speculation is needed.

4. Considering figures 5 and 6, there must be a dose effect of gentamicin bewteen 300 and 600 uM.   However, fcell dealth rate in Figure 2 did not differ between 150 and 600 uM doses. How could the authors eleborate on this phenomenon. Is there a discrepancy between overall and apoptic cell deaths? It feels like apoptosis or mitophagy-related cell death could have a partial contribution to cell death. Discussion is required.

5.  In discussion, the authors pointed out HEI-OC1 is more resistant to gentamicin than OPC. I wonder if HEI-OC1 was also tested in this study. It would be better to provide the data together in Figure 3a if so.

6.  In lines 514-518, it can be explained by the absence of MET channels, which may hinder the accumulation of gentamicin. It is consistent that the hair cells of the mice within P2-3 of age does not have MET activity and is resistent to the treatment of aminoglycoside, which can not be used for induction of drug-induced hearing loss. More in-depth discussion would be helpful to understand the outcomes.  

Reviewer 2 Report

This is a well-written manuscript describing an in vitro oxidative stress model of the human inner ear using otic progenitor cells (OPCs) derived from human induced pluripotent stem cells (hiPSCs). Cells were exposed to hydrogen peroxide or ototoxic drugs (gentamicin and cisplatin) that induce oxidative stress to evaluate subsequent cell viability, cell death, reactive oxygen species (ROS) production, mitochondrial activity, and apoptosis (caspase 3/7 activity). The results show OPC cell vulnerability to oxidative stress, suggesting an effective OPC-based model to simulate oxidative stress conditions in the human inner ear. The manuscript provided very detailed materials and methods in cell culture, stress exposure, cell array imaging and statistical analysis, which is a big plus for the reproducibility of experiments by others. I only have few minor comments to help further improve the manuscript.

 Please clarify if this study requires any ethical protocols, and if so please state the protocol details.

Figure 3a shows OPC viability was significantly reduced starting at 150 μM gentamycin, while Figure 5b shows caspase 3/7 enzyme activity didn’t change much in the 300 μM gentamicin group, which is even higher than 150 μM gentamycin, why is that? Same question for the OPC viability significantly reduced starting at 12.5 μM cisplatin in Fig 4a, yet not much change in caspase 3/7 enzyme activity even in the 25μM cisplatin group in Fig 5d.

In discussion, do you think “organ-on-a-chip” technology could better model the inner ear ex vivo?

There is typo Line 169, "from power" should be "from powder"

Round 2

Reviewer 1 Report

All responses addressed my concerns well.

It's suitable for the publication